# Early-Life Exposure to Traffic-Related Air Pollutants Induced Anxiety-like Behaviors in Rats via Neurotransmitters and Neurotrophic Factors

**DOI:** 10.3390/ijms24010586

**Published:** 2022-12-29

**Authors:** Chaw Kyi-Tha-Thu, Yuji Fujitani, Seishiro Hirano, Tin-Tin Win-Shwe

**Affiliations:** 1Department of Immunology, School of Medicine, International University of Health and Welfare, 4-3, Kozunomori, Narita 286-8686, Chiba, Japan; 2Health and Environmental Risk Division, National Institute for Environmental Studies, Tsukuba 305-8506, Ibaraki, Japan

**Keywords:** anxiety, air pollutants, molecular markers, neurotransmitters, rats

## Abstract

Recent epidemiological studies have reported significantly increasing hospital admission rates for mental disorders such as anxiety and depression, not only in adults but also in children and adolescents, indicating more research is needed for evaluation of the etiology and possible reduction and prevention of these disorders. The aim of the present study was to examine the associations between perinatal exposure to traffic-related air pollutants and anxiety-like behaviors and alterations in neurological and immunological markers in adulthood using a rat model. Sprague Dawley pregnant rats were exposed to clean air (control), diesel exhaust (DE) 101 ± 9 μg/m^3^ or diesel exhaust origin secondary organic aerosol (DE-SOA) 118 ± 23 μg/m^3^ from gestational day 14 to postnatal day 21. Anxiety-related behavioral tests including open field tests, elevated plus maze, light/dark transition tests and novelty-induced hypophagia were performed on 10-week-old rats. The hippocampal expression of neurotransmitters, neurotrophic factors, and inflammatory molecular markers was examined by real-time RT-PCR. Anxiety-like behaviors were observed in both male and female rat offspring exposed to DE or DE-SOA. Moreover, serotonin receptor (*5HT1A*), dopamine receptor (*Drd2*), brain-derived neurotrophic factor and vascular endothelial growth factor A mRNAs were significantly decreased, whereas interleukin-1β, cyclooxygenase-2, heme oxygenase-1 mRNAs and microglial activation were significantly increased in both male and female rats. These findings indicate that brain developmental period exposure to traffic-related air pollutants may induce anxiety-like behaviors via modulation of neurotransmitters, neurotrophic factors, and immunological molecular markers, triggering neuroinflammation and microglia activation in rats.

## 1. Introduction

Air pollution is has emerged as a global health problem, especially in rapidly developing countries. The World Health Organization estimates that over four million premature deaths in every year are due to air pollution. Particulate matter (PM)_2.5_ consists of diesel exhaust particles (DEP), which are major precursors of secondary organic aerosol (SOA) formation. Using commercial nano-sized carbon black and whole-body exposure chambers established in our research institute, we have reported the effect of exposure to air pollutants on alteration of brain neurotransmitters [1,2] and impaired behavior such as spatial learning and novel object recognition ability in mice and rats [3,4,5]. In addition, impaired social behavior and altered hypothalamic expression of social behavior-related genes have been observed in adult male mice offspring exposed to DE-SOA during the gestational and lactational periods [6].

In human studies, cognitive deficit, poor olfactory and auditory functions, anxiety, depression and other mental disorders have been reported [7,8,9,10]. EEG changes were found in subjects exposed to DE (300 μg/m^3^) for 1 h [11]. In addition, post-mortem examination of individuals exposed to high levels of air pollutants have shown increased oxidative stress markers and neuroinflammation [12,13]. A recent epidemiological study has demonstrated that short-term exposure to traffic-related pollutants, especially PM_2.5_, was significantly associated with hospital admissions for anxiety in the coastal city of Qingdao, China [14]. In that study, it was reported that females and younger individuals were more susceptible than males and elderly individuals.

In animal studies, it was reported that particles or gases in air pollutants can enter the brain by directly through the nasal olfactory mucosa [15,16], and by entering circulation and then cross the blood brain barrier [17,18]. Significantly increased proinflammatory cytokine tumor necrosis factor-α and microglia activation were found in mice exposed to DE compared with those exposed to filtered air [19]. Using the BrdU/NeuN co-localization method, Costa et al. showed that DE exposure significantly decreases neurogenesis in the hippocampal subgranular and subventricular zones [20]. Disruption of DNA methylation in the brain was observed in mice exposed to DE prenatally, by affecting genes involved in neuronal differentiation and neurogenesis [21]. In addition, using DE and DE-SOA as models of air pollutants, we have shown that perinatal exposure to DE-SOA induces autism-like behavior and dysregulation of neuroimmune responses in rats [22].

The hippocampus is the important brain area for cognitive functions such as episodic memory and spatial navigation. In addition, it is also involved in the pathogenesis of mood and anxiety disorders. The dorsal hippocampus contributes to cognitive functions, and the ventral hippocampus modulates emotional regulation [23,24]. The hippocampus, amygdala and prefrontal cortex are the brain areas which play major roles in emotional behaviors and functions. The neural connections projecting from ventral hippocampus to the prefrontal cortex are unidirectional. The ventral hippocampus also has bidirectional connections with the amygdala, and the amygdala has bidirectional connections with the prefrontal cortex. Because the amygdala and hippocampus are known to be involved in emotional and contextual memory processing, stress likely contributes to the dysregulation of these functions. Using freely moving calcium imaging and optogenetics, it was reported that optogenetic activation of ventral hippocampal terminals in the lateral hypothalamus, but not the basal amygdala, increased anxiety and avoidance. The ventral hippocampal area is enriched in anxiety cells; thus, the hippocampus can rapidly influence innate anxiety behavior directly via the ventral hippocampus–lateral hypothalamus pathway [25]. Transgenic animals with impaired hippocampal neurogenesis exhibit significantly increased anxiety-like behaviors [26]. Taken together, these studies indicate that the hippocampus is the essential brain region for anxiety-like behaviors that result from exposure to environmental insults. Thus, in the present study, the hippocampus was selected to examine environmental pollutant-induced anxiety-like behaviors in rats.

Currently, prevalence of mental disorders such as anxiety and depression is increasing. However, little is known about the association between environmental pollutant-exposure and anxiety. This prompted us to address this gap in the research concerning whether there is an association between developmental exposure to DE or DE-SOA and anxiety.

## 2. Results

### 2.1. Assessment of Overall Toxicity

At the time of sample collection, body weight and brain weight were measured in 10-week-old male and female rat offspring of the control and exposure groups to detect the overall toxicity. Body weight was not different between groups in both male and female rat offspring. However, brain weight was significantly increased in DE-SOA groups of both male and female rat offspring (*p* < 0.01, *p* < 0.05 vs. corresponding control, Figure 1).

### 2.2. Anxiety-Related Behavior Assessment

#### 2.2.1. Open Field Test

The OFT is an experimental test used to determine general locomotor activity levels, anxiety, and exploratory behavior of rodents in scientific research. The male and female rats exposed to DE and DE-SOA groups showed a significantly decreased number of entries to the center and time stay in the center compared with the control group (*p* < 0.01; *p* < 0.05, Figure 2). Locomotor activity or distance traveled was not different between the control and exposure groups. However, grooming, rearing and defecation were increased in DE or DE-SOA exposed male and female rats. Our findings indicate that DE or DE-SOA exposure during the perinatal period may induce emotional insecurity and restlessness in male and female rats.

#### 2.2.2. Elevated Plus Maze

The elevated plus maze (EPM) test was performed to assess the emotional level of the rats and rodents. The emotional security level was calculated based on two factors of entering the open arms and time spent in the arms. The male and female rats exposed to DE or DE-SOA groups showed a significantly decreased open arms entry compared with the control group (*p* < 0.01, *p* < 0.05, Figure 3) but no significant difference in time spent in open arms. In the EPM test, a decreased number of open arm entries, and reduced time spent in the open arms indicate increased anxiety responses.

#### 2.2.3. Light Dark Transition Test

In the light dark transition (LDT) test, the number of entries and time spent in the light compartment were not different between the control and exposure groups in male rats. However, DE-exposed female rats showed an increased number of entries and time spent in the light compartment compared with the control rats (*p* < 0.01, *p* < 0.05, Figure 4). These findings indicate that DE exposure may affect some part of anxiety behavior in a sex-specific manner.

#### 2.2.4. Novelty-Induced Hypophagia Test

Novelty-induced hypophagia (NIH) is a validated test for anxiety-like behavior induced by a new environment [27,28]. Latency to approach and latency to eat sweet potato were decreased in Day 2 and Day 3 compared with Day 1 in all groups. On novelty day, latency to approach and latency to eat sweet potato were delayed in DE- or DE-SOA-exposed male and female rats compared with the control group (*p* < 0.05, Figure 5). These findings indicate that perinatal exposure to DE or DE-SOA may induce anxiety-like behavior in male and female rats in new or novel environments.

### 2.3. Messenger RNA Expression Assay

#### 2.3.1. Neurotransmitter Receptor Levels in the Hippocampus

Messenger RNA expression level of neurotransmitters such as serotonin receptor *5HT1A* and dopamine receptor *Drd2* mRNA in the hippocampus were examined. In male rats, *5HT1A* and *Drd2* mRNAs were significantly decreased in the DE- and DE-SOA-exposed groups compared with the control group (*p* < 0.01, *p* < 0.05 vs. control, Figure 6A). In contrast, in female rats, *5HT1A* and *Drd2* mRNAs were significantly decreased in the DE-SOA-exposed group only compared with the control group (*p* < 0.01, Figure 6B). These findings indicate that perinatal exposure to DE or DE-SOA may induce alteration in neurotransmitters in male and female rats in a sex-dependent manner.

#### 2.3.2. Neurotrophin Levels in the Hippocampus

Brain-derived neurotrophic factor (*BDNF*) and vascular endothelial growth factor A (*VEGFA*) in the hippocampus were examined. In male rats, both *BDNF* and *VEGFA* mRNAs were significantly decreased in the DE-SOA-exposed group but not in the DE group compared with the control group (*p* < 0.05 vs. control, Figure 7A). However, in female rats, *BDNF* mRNA was significantly decreased in the DE-SOA-exposed group compared with the control group (*p* < 0.01 vs. control, Figure 7B). *VEGFA* mRNA in the female rats did not show significant difference between the control and exposure groups. These findings indicate that perinatal exposure to DE-SOA may induce reductions in *BDNF* levels in male and female rats.

#### 2.3.3. Proinflammatory Cytokines and Oxidative Stress Markers in the Hippocampus

To detect DE or DE-SOA-induced inflammation in the hippocampus, the inflammatory molecular markers such as interleukin *(IL)-1β*, cyclooxygenase *(COX)2* and heme oxygenase *(HO)-1* in the hippocampus were examined. *IL-1β*, *COX2* and *HO1* mRNAs were significantly increased in the DE-SOA-exposed male rats compared with the control group (*p* < 0.05; Figure 8A). However, the expression levels of *IL-1β* mRNA only was remarkably increased in female rats exposed to DE-SOA compared with the control group (*p* < 0.05; Figure 8B). These results indicate that perinatal exposure to DE or DE-SOA may induce inflammatory responses in male and female rats in a sex-dependent manner.

### 2.4. Immunohistochemical Analyses

Microglial marker *Iba1* was used to examine the activation of microglia in the hippocampus. Representative digital photomicrographs of *Iba1*-immunostained sections taken from the hippocampus of the control, DE- or DE-SOA-exposed groups are shown in Figure 9. Microglial activation was markedly increased in the hippocampus of the De- and DE-SOA-exposed group compared with that in the control group. *Iba1*-positive microglia in the DG area of the hippocampus were quantified under a high-power field using ImageJ software. 

## 3. Discussion

The major findings of the present study were that the gestational and lactational exposure to DE-SOA induces the following responses: (1) anxiety-like behaviors such as decreased entry to the center and time stay in the center in OFT, decreased open arm entry and open arm time in EPM tests, and latency to approach and latency to eat food on the novelty day in NIH tests; (2) downregulation of anxiety-related gene serotonin receptor *5HT1A* and neurotrophic factor *BDNF*; (3) upregulation of proinflammatory cytokine *IL-1β*, *COX2* and oxidative stress marker *HO-1*; and (4) increased microglia activation in the hippocampus in male and female rats. Our findings suggest that the perinatal exposure to DE-SOA induces anxiety-like behavior in rats by triggering neuroinflammation and neuropsychological disorders via neurological and immunological biomarkers in the brain.

According to the WHO, air pollution is the fourth largest risk factor for global mortality [29]. Major air pollutants are organic hydrocarbons, particulate matter (PM), carbon monoxide (CO), nitrogen dioxide (NO_2_), carbon dioxide (CO_2_), and heavy metals [30]. The major sources of these air pollutants and heavy metals are from transportation [31]. Thus, traffic-related air pollution is a major contributor to global air pollution. Air pollution due to gasoline and diesel emissions from internal combustion engines of automobiles, trucks, locomotives, and ships leads to 800,000 premature deaths annually due to pulmonary, cardiovascular, and neurological complications [32]. People living and working in areas of heavy vehicle traffic have high susceptibility to mental health problems and learning and memory impairment such as anxiety, depression, and cognitive deficits.

Using animal studies, we have reported that exposure to diesel exhaust and diesel engine derived secondary organic aerosols is associated with adverse effects in the central nervous system (CNS) via increased oxidative stress and neuroinflammation. We have shown that the effect of exposure to the nano-sized carbon black (14 nm) and nanoparticle-rich diesel exhaust particles causes alteration of brain neurotransmitters [1,2] and impairs behavior such as spatial learning and novel object recognition ability [3,4,5] in mice and rats. In addition, impaired social behavior and altered hypothalamic expression of social behavior-related genes were observed in adult male mice offspring exposed to DE-SOA during the gestational and lactational periods [6]. In addition, using DE and DE-SOA as models of air pollutants, we have shown that perinatal exposure to DE-SOA induces autism-like behavior and dysregulation of neuroimmune responses in rats [22]. However, little is known about the association between environmental pollutant-induced autism and anxiety. This prompted us to address a research gap concerning the associations between developmental exposure to DE and/or DE-SOA and anxiety.

In preliminary studies, our research group investigated the critical period of exposure to nano-sized diesel exhaust particles for brain functions and behaviors in relation to different phases of developmental periods such as the whole gestational period, early, middle, and late gestational period, neonatal period, and perinatal period. We found that the perinatal period (GD 14 to PND21) was the most vulnerable period for the whole-body exposure to DE or DE-SOA. In addition, epidemiological studies have reported that exposure to air pollutants during prenatal and childhood periods is associated with neurodevelopmental and emotional disorders including ASD, ADHD, anxiety and depression [33,34,35]. Thus, in this study, we have selected the perinatal period to examine the effect of developmental exposure to DE or DE-SOA on neuropsychiatric diseases, especially anxiety-like behavior and neurological and immunological markers, using rat models.

In this study, four types of anxiety-like behavior tests including OFT, LDT test, EPM test and NIH tests were performed. In OFT, the center entry number and time spent in the center were significantly reduced in both male and female rats exposed to DE or DE-SOA. However, in the LDT test, only female rats exposed to DE showed increased entry number and time spent in the light compartment compared with the control group. In the EPM test, the male and female rats exposed DE and DE-SOA groups showed a significant decrease in the ability to enter the open arms, and shorter time spent in the open arms compared with the control group. Moreover, latency to approach and latency to eat in a novel environment were reduced significantly in both male and female rats exposed to DE or DE-SOA perinatally. Taken together, these results indicate that the male and female rats exposed to DE or DE-SOA during the perinatal period had induced anxiety-like behaviors.

The hypothalamic–pituitary–adrenal (HPA) axis and the serotonergic neuromodulatory system plays a crucial role in mediating anxiety behaviors [36]. 5-HT receptor abnormalities can induce a deficiency in brain serotoninergic activity, which causes increased vulnerability to anxiety. Among the 5HT receptors, it was reported that the *5-HT1A* receptor knockout mice show increased anxiety [37,38], and in contrast, the 5-HT1A receptor overexpressed mice show decreased anxiety [39]. Although we did not detect the plasma corticosteroid levels in the present study, exposure to DE or DE-SOA can lead to the release of corticosteroids from the adrenal gland via the HPA axis, which can influence postsynatptic *5-HT1A* function.

In human and animal studies, it was reported that dopamine D2 receptors play a major role in the pathogenesis of anxiety and depression [40,41,42]. In addition, *D2Rs* regulate anxiety-related behavior and elevate plus-maze-associative memory via the interactions of the dorsal hippocampus and the nucleus accumbens dopaminergic system [43]. In the present study, dopamine receptor Drd2, mRNAs in the hippocampus were remarkably reduced in DE-SOA-exposed male and female rats.

The vascular endothelial growth factor *(VEGF)-A* is essential for neurons because they require a large vascular supply and insufficient expression of *VEGFA* will lead to defects in the brain vascularization and neuronal apoptosis. *VEGFA* not only induces vascular permeability in tissues, including the blood brain barrier, but also facilitates the neurogenesis and proliferation of neurons, influences synaptic plasticity and modulates synaptic transmission in the adult hippocampus [44]. Moreover, *VEGF* can act as a neuroprotective factor in the adult brain, inhibiting apoptosis and inducing the growth of vascular and neuronal networks. Stress can decrease the hippocampal expression of *VEGF* [45]. In this study, *VEGFA* mRNAs in the hippocampus were significantly reduced in DE-SOA-exposed males, but not in female rats.

Brain-derived neurotrophic factor is a neurotrophin that influences cell growth, cell differentiation, and synaptic modification [46,47] and is highly expressed in the developing and adult hippocampus [48,49]. Recently, *Val66Met*, a single nucleotide polymorphism of the coding region of the *BDNF* gene has been identified as a risk factor for anxiety disorders and post-traumatic stress disorder [50,51]. In the present study, *BDNF* and *VEGFA* mRNAs in the hippocampus were remarkably reduced in DE-SOA-exposed male rats, and *BDNF* mRNA was remarkably reduced in DE-SOA-exposed male and female rats perinatally.

Microglia are natural immune cells in the brain and play a key role in neuroinflammation. Microglia detect changes in the surroundings and provide immunosurveillance activity. The functions of the microglia are maintaining neuronal synapses, identifying pathogens, and removing cellular debris. In this study, microglia marker *Iba1* immunoreactivity was significantly increased in the dentate gyrus of the hippocampus of DE- or DE-SOA-exposed groups of male and female rats. This microglial activation was more remarkable in males compared with female rats.

In this study, some sex-dependent anxiety-like behavior (Table 1) and expression of molecular markers (Table 2) were observed. The exact etiology is not known, but females may possess additional protective activities compared with males. It has been reported that estrogen exerts protective actions by suppressing the neurotoxic stimulus or increasing the resilience of the brain to a given injury [52]. Moreover, paraoxonase-2 is a mitochondrial enzyme, possesses potent antioxidant and anti-inflammatory properties, and its expression in female monkeys was significantly higher than in male monkeys in multiple brain regions [53].

Significant differences in anxiety-like behaviors and neurological and immunological marker mRNAs between the control and DE-SOA groups were observed, but not between the DE and DE-SOA groups. Taken together, these results indicate that developmental exposure to DE or DE-SOA induces disturbances in the neurotransmitter system and neurotrophin production and also induces neuroinflammation via activated microglia. These release inflammatory molecules such as cytokines and oxidative stress markers, which may lead to anxiety-related behaviors in DE- or DE-SOA-exposed rats.

## 4. Materials and Methods

### 4.1. Animals

Twenty-four pregnant Sprague Dawley rats (gestational day (GD) 8) were purchased from Oriental Yeast Co., Ltd. (Tokyo, Japan) and exposed to clean air (control, *n* = 8), DE (*n* = 8), and DE-SOA (*n* = 8) from GD 14 to postnatal day (PND) 21 in whole-body exposure chambers. Food and water were given ad libitum. Date of birth was recorded as PND 0 and the offspring were housed in cages with dams under controlled environmental conditions (temperature, 22 ± 0.5 °C; humidity, 50 ± 5%; lights on 07:00–19:00 h). The pregnant rats were exposed clean air, DE or DE-SOA for 5 h per day (from 10:00 p.m. to 3:00 a.m.), 5 days a week excluding weekends from gestational day 14 to postnatal day 21 with their pups (Figure 10). The numbers of pups born were 253 (116 male and 137 female) in the control, 130 (68 male and 62 female) in the DE group, and 135 (63 male and 72 female) in the DE-SOA group. We used 3 male and 3 female pups from each dam (total 72 male and 72 female pups). Among them, 10 male and 10 female rats from 3 groups (total 60) were used for the anxiety-like behavior tests, and 5 male and 5 female rats from 3 groups (total 15) were used for immunohistochemical analyses.

The pups were weaned at PND 21 and 3~4 pups of the same sex were housed in a plastic cage. Anxiety-like behavior tests were performed at 10 weeks old. Behavioral testing was performed between 09:00 and 13:00 h. Before performing each test, the apparatus to be used was cleaned with 70% ethanol. After completing the anxiety behavioral tests, the rats were sacrificed under deep pentobarbital anesthesia. The hippocampus was collected from each animal, frozen quickly in liquid nitrogen, then stored at −80 °C until extraction of the total RNA. The experimental protocols were approved by the Ethics Committee of the Animal Care and Experimentation Council of the National Institute for Environmental Studies (NIES), Japan (AE-20-05, 9 March 2020). All efforts were made to minimize the number of animals used and their suffering.

### 4.2. Exposure to Clean Air, DE, or DE-SOA

The whole-body inhalation exposure chambers were established at the National Institute for Environmental Studies, Japan, as described previously [54]. Briefly, an 81-diesel engine (J08C; Hino Motors Ltd., Hino, Japan) was used to generate diesel exhaust. The engine was operated under a steady-state condition for 5 h a day. Our driving condition of the diesel engine was not simulated to any special condition in the real world. The engine operating condition (2000 rpm engine speed and 0 Nm engine torque) promoted the generation of high concentrations of nano-size particles. There were three chambers: a control chamber receiving clean air filtered through a high efficiency particulate air (HEPA) filter and a charcoal filter (referred to as “clean air”), the diluted exhaust (DE) chamber, which was without mixing of O_3_, and the DE-SOA chamber, which was generated by mixing DE with ozone at 0.6 ppm after secondary dilution. The secondary dilution ratios in the DE and DE-SOA chambers were the same, which resulted in the same particle and gaseous concentrations when O_3_ was not mixed. In fact, the concentrations of particles in DE-SOA were higher when O_3_ was mixed, and the concentrations of DE and DE-SOA were 101 ± 9 μg/m^3^ and 118 ± 23 μg/m^3^, respectively. The increased mass concentration was due to the generation of secondary particles. The temperature and relative humidity inside each chamber were adjusted to approximately 22 ± 0.5 °C and 50 ± 5%, respectively. In detail, sample air was taken from the inhalation chamber (2.25 m^3^) using stainless steel tubing. The gas concentrations (CO, CO_2_, NO, NO_2_, and SO_2_) were monitored using a gas analyzer (Horiba, Kyoto, Japan) (Table 3). The CO and NO_x_ concentrations in both chambers were similar, but the NO and NO_2_ concentrations differed because NO was oxidized to NO_2_ by reaction with O_3_. The particle size distributions were measured using a scanning mobility particle sizer (SMPS 3034, TSI Instruments, Shoreview, MN, USA). The modal sizes of the particles used in the present study were 22.69 ± 1.47 nm for DE and 24.45 ± 1.21 nm for DE-SOA. The particles were collected using a Teflon filter (FP-500; Sumitomo Electric, Osaka, Japan) and a Quartz fiber filter (2500 QAT-UP; Pall, Pine Bush, NY, USA), and the particle mass concentrations were measured using a Teflon filter. The particle weights were measured using an electrical microbalance (UMX 2; Mettler-Toledo, Columbus, OH, USA; readability 0.1 μg) in an air-conditioned chamber (CHAM-1000; Horiba) under constant temperature and relative humidity conditions (21.5 °C, 35%). For the Quartz fiber filter, the quantities of elemental carbon (EC) and organic carbon (OC) were determined using a carbon analyzer (Desert Research Institute, Reno, NV, USA). The EC to OC ratios in the present study were 0.15 ± 0.06 for the control chamber, 0.36 ± 0.03 for the DE chamber, and 0.38 ± 0.03 for the DE-SOA exposure chamber.

### 4.3. Behavioral Assessment

All the animals were kept in the animal experimental room next to the behavior test room. The animals were placed in the testing room 30 min before the behavior test on the experimental day. Apart from the tested animal, other animals were kept in the animal experimental room during the whole experiment period. For every behavior test, a sufficient interval between test rats was maintained, and after cleaning with 70% ethanol, a hand-held fan was used to remove the smell of ethanol.

#### 4.3.1. Open Field Test

The open field test is validated for use in the measurement of anxiety-related behaviors in rodents. The apparatus consisted of a square box (100 cm width × 100 cm length × 35 cm height) made of clear plastic. Inside the box, a center square (25 cm × 25 cm) was outlined in red tape. The apparatus was placed on the floor. Before and between trials, the box was cleaned with 70% ethanol and dried off. The rats were placed in the arena and allowed to explore it for 5 min. The movements were recorded by a video tracking system and stored on a computer. The video recordings were later scored for time spent in the center square, time spent moving throughout the box (locomotor time or distance travel), the number of entrances into the center, grooming (washing the face and body), rearing (standing on the hind limbs) and defecation (fecal pellets).

#### 4.3.2. Elevated Plus Maze Test

The EPM apparatus was purchased from Muromachi Kikai (Tokyo, Japan) and consisted of two open and two closed arms. It was elevated 50.0 cm above the floor and each arm was 48.3 cm in length and 12.7 cm in width. The walls of the closed arms were 29.2 cm tall. On test day, the animals were moved to testing room 30 min before the experiment. At the beginning of testing, each rat was placed at the center of the crossing arms, facing the open arm and allowed to move freely within the maze for 5 min. Between trials, the apparatus was cleaned with 70% ethanol. Time spent in the open and closed arms, and entries into the open and closed arms were recorded. An animal was recorded as entering an arm when all four paws were placed in that arm. The illumination at the entrance into the open arms was set to ~90 lux.

#### 4.3.3. Light Dark Transition Test

The apparatus contained two compartments of white and black opaque plexiglas (34 × 24 × 24 cm), with a rectangular aperture (8 cm diameter) between the two chambers permitting free movement between the chambers (Muromachi Kikai, Japan). The white compartment was named “the light compartment” and the black compartment was named “the dark compartment”. A black, non-translucent lid covered the dark compartment to exclude ambient room light. The light compartment was covered by a transparent plastic lid, which allowed room light into the apparatus. To begin testing, an animal was placed in the light compartment, facing away from the aperture between the chambers. The animal was allowed to investigate the chambers for 5 min and the number of entries and time spent in each compartment was recorded by video.

#### 4.3.4. Novelty-Induced Hypophagia Test

The novelty-induced hypophagia (NIH) is a validated test for anxiety-like behavior induced by new environment [13,14]. Training Phase: Rats were removed from their home-cages and isolated for 30 min in a new standard cage (“training cage”), which was lined with fresh bedding and provided access to food and water, to acclimate to the novel environment (animals were not food or water restricted at any point during NIH testing). After the 30 min isolation period, a sweet potato slice was placed in the cage. The animal’s approaches to the sweet potato slice were recorded by video-assisted Any-maze software for 15 min. After 15 min, the rats were returned to their home-cages. The sweet potato slices were weighed before and after the training phase to determine how much of the sweet potato slice had been consumed by the rat. This process was repeated for the next 3 consecutive days using the same training cage.

Novelty Test: On the fourth day, rats were allowed to acclimate to their “training cage” for 30 min, as described previously for the training phase. Following this acclimation period, the animal was transferred to a novel plastic tub (58.42 cm (L) × 43.18 cm (H) × 31.75 cm (W) with no bedding/water/food) and a sweet potato slice was immediately placed in the novel tub. Explorations of the sweet potato slice in the novel environment were recorded by video for 15 min. After 15 min, the rats were returned to their home-cages. Sweet potato slices were weighed before and after the novelty test to determine the amount of consumption, as the amount of sweet potato slice consumed has also previously been reported as a measure of anxiety-like behavior [13]. Experimental videos were later analyzed for the time taken by the animal to approach and begin to consume the sweet potato (latency), as well as the amount of sweet potato consumed.

### 4.4. MessengerRNA Expression Assay

After completion of all anxiety-related behavioral tests, 10-week-old male and female rats (*n* = 10 from each group) were sacrificed under deep pentobarbital anesthesia and the hippocampus was collected from each group for mRNA analyses. Briefly, the total RNA was extracted from the hippocampal samples using the BioRobot EZ-1 and EZ-1 RNA tissue mini kits (Qiagen GmbH, Hilden, Germany). Then, the purity and quality of the total RNA were examined using the ND-1000 NanoDrop RNA Assay protocol (NanoDrop, Wilmington, DE, USA), as described previously [6]. Next, first-strand cDNA synthesis from the total RNA was made using SuperScript RNase H−Reverse Transcriptase II (Invitrogen, Carlsbad, CA, USA), according to the manufacturer’s protocol. The mRNA expression levels were examined by RT-PCR (Light Cycler 96, Roche, Mannheim, Germany). The tissue 18S rRNA level was used as an internal control. The following primer sequences were used in the present study. Some primers (*BDNF*, NM_012513; *IL-1β*, NM_008361; *COX2*, NM_011198; and *HO-1*, NM_010442) were purchased from Qiagen, Sample and Assay Technologies. Other primers were purchased from Hokkaido System Science (Sapporo, Japan) as follows: 18S (forward 5′-TACCACATCCAAAAGGCAG-3′, reverse 5′-TGCCCTCCAATGGATCCTC-3′), TNFα (forward 5′-GGTTCCTTTGTGGCACTTG-3′, reverse 5′-TTCTCTTGGTGACCGGGAG-3′), 5HT1A (forward 5′-CTGGGGACGCTCATTTTCT-3′, reverse 5′-CCAAGGAGCCGATGAGATAG-3′), Drd2 (forward 5′-TGTACAATACGCGCTACAGCTCCA-3′, reverse 5′-ATGCACTGCGTTCTGGTCTGCGTTA-3′), and VEGFA (forward 5′-CTGCTGTAACGATGAAGCCCTG-3′, reverse 5′-GCTGTAGGAAGCTCATCTCTCC-3′). Data were analyzed using the comparative threshold cycle method. Then, the relative mRNA expression levels were expressed as mRNA signals per unit of 18S rRNA expression.

### 4.5. Immunohistochemical Analyses

In the brain, microglia play a role as major immune cells, and microglial activation indicates neurotoxicity. The hippocampal tissue sections were immunostained with microglial marker *Iba1*. Briefly, the brain sections were immersed in absolute ethanol followed by 10% H_2_O_2_ for 10 min each at room temperature. After rinsing in 0.01-M phosphate buffer saline, the sections were blocked with 2% normal swine serum in PBS for 30 min at room temperature and then reacted with goat polyclonal anti-*Iba1* (diluted 1:100; abcam: ab5076; Tokyo, Japan) in PBS for 1 h at 37 °C. Then, the sections were reacted with biotinylated donkey anti-rabbit IgG (1:300 Histofine; Nichirei Bioscience, Tokyo, Japan) in PBS for 1 h at 37 °C. The sections were then incubated with peroxidase-tagged streptavidin (1:300, ABC KIT) containing PBS for 1 h at room temperature. After a further rinsed in PBS, *Iba1* immunoreactivity was detected using a Dako DAB Plus Liquid System (Dako Corp., Carpinteria, CA, USA). To detect the immunoreactivity of *Iba1* in the hippocampus, photomicrographic digital images (150 dpi, 256 scales) of the hippocampal regions were taken using a CCD camera connected to a light microscope. Numbers of *Iba1*-positive microglia in the hippocampus were quantified in high power field using ImageJ software (5 fields per section and 3 sections per mouse, *n* = 5 rats/group).

### 4.6. Statistical Analysis

All the data were expressed as the mean ± standard error (SE). The statistical analyses were performed using the StatMate II statistical analysis system for Microsoft Excel, Version 5.0 (Nankodo Inc., Tokyo, Japan). The data were analyzed using a one-way analysis of variance with a post hoc analysis using the Bonferroni/Dunn method. Differences were considered significant at *p* < 0.05.

## 5. Conclusions

At least in part, anxiety-like behaviors were observed in both male and female rat offspring exposed to DE or DE-SOA. Moreover, hippocampal expression of neurotransmitters, including serotonin receptor *5HT1A*, dopamine receptor *Dr2d*, *BDNF* and *VEGFA* mRNAs, were significantly decreased in both male and female rats. On the other hand, inflammatory markers, including *IL-1β* and *COX2*, oxidative stress marker *HO1* and microglia activation in the hippocampus, were significantly increased in both male and female rat offspring exposed to DE or DE-SOA. Activated microglia were prominent in the hippocampus of both male and female rat offspring exposed to DE or DE-SOA during the perinatal period. Our results indicate that developmental exposure to air pollutants induces anxiety-like behaviors via modulation of neurotransmitters and neurotrophic factors in rats.

The differences between DE and DE-SOA are complex, as shown in Table 1 and Table 2. The effects of DE or DE-SOA are brain region-specific, and it would be preferable to examine the related brain regions simultaneously. Hippocampal volume, neurogenesis, apoptosis and autophagy are other mechanisms that contribute to hippocampus-dependent anxiety-like behaviors following air-pollutant exposure. The impact of exposure to DE-SOA appears greater than DE but is not statistically significant. DE is a precursor of DE-SOA, which contains ozone oxidized to form DE-SOA. Moreover, we could not determine exactly the differences between DE and DE-SOA in translocation to the brain or their effects on the anxiety-like behavior and molecular markers in the rat model. The limitation of this study was that only the hippocampus was examined. We plan to examine the constituents of DE or DE-SOA and their translocation to the brain and the different effects on other behaviors and molecular markers in future studies.

## Figures and Tables

**Figure 1 ijms-24-00586-f001:**
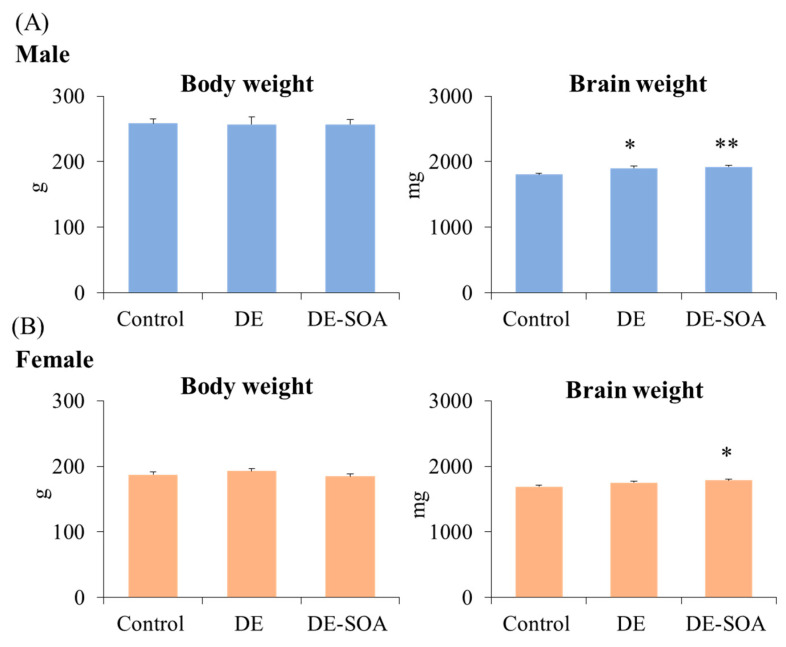
Assessment of overall toxicity. Body weight and brain weight in 10-week-old (**A**) male and (**B**) female rats of the control, DE- or DE-SOA-exposed groups (*n* = 10, ** *p* < 0.01, * *p* < 0.05 vs. corresponding control).

**Figure 2 ijms-24-00586-f002:**
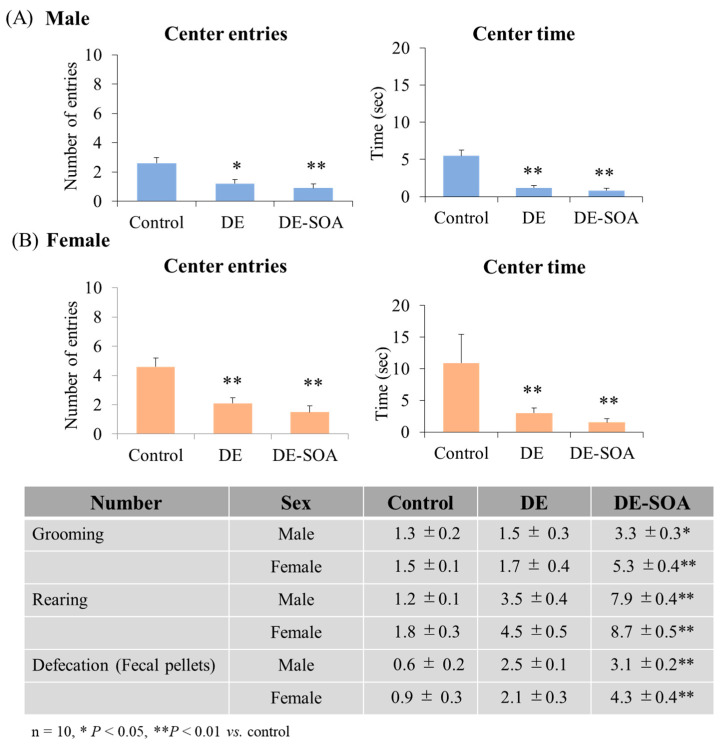
Open field test. Number of entries to the center and time stay in the center in 10-week-old (**A**) male and (**B**) female rats of the control and the DE- or DE-SOA-exposed groups (*n* = 10, ** *p* < 0.01, * *p* < 0.05 vs. control). Table shows the number of grooming, rearing and defecation (fecal pellets) during open field test.

**Figure 3 ijms-24-00586-f003:**
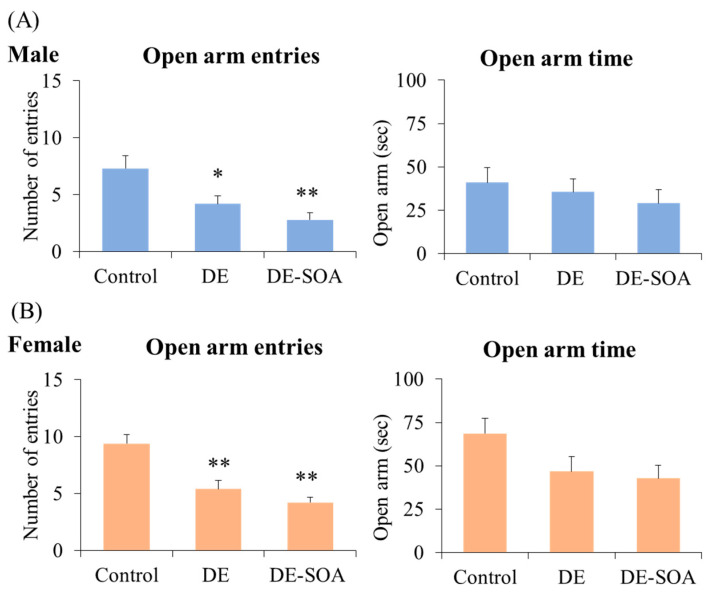
Elevated plus maze test. Number of entries to the open arm and time spent in the open arm in 10-week-old (**A**) male and (**B**) female rats of the control and the DE- or DE-SOA-exposed groups (*n* = 10, ** *p* < 0.01, * *p* < 0.05 vs. corresponding control).

**Figure 4 ijms-24-00586-f004:**
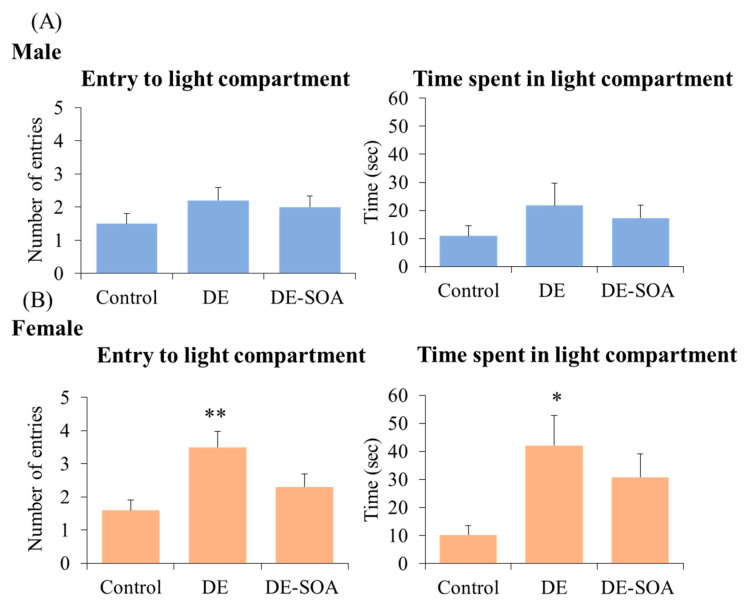
Light dark transition test. Number of entries and time spent in the light compartment in 10-week-old (**A**) male and (**B**) female rats of the control and the DE- or DE-SOA-exposed groups (*n* = 10, ** *p* < 0.01, * *p* < 0.05 vs. corresponding control).

**Figure 5 ijms-24-00586-f005:**
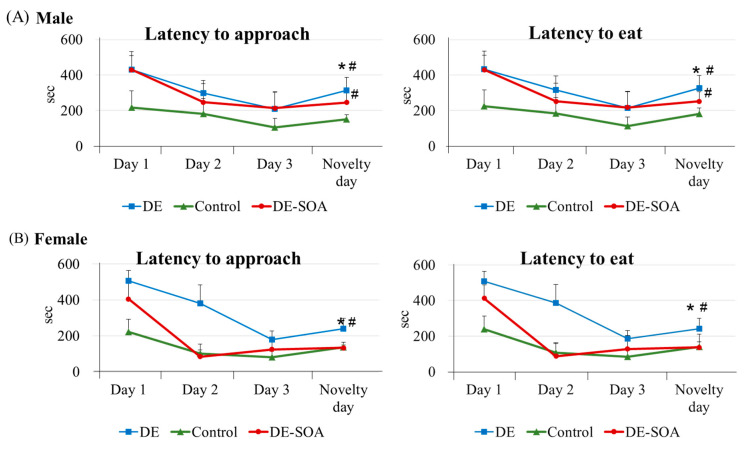
Novelty-induced hypophagia test. Latency to approach and latency to eat in 10-week-old (**A**) male and (**B**) female rats of the control and the DE- or DE-SOA-exposed groups (*n* = 10, * *p* < 0.05 vs. corresponding Day 3 and # *p* < 0.05 vs. corresponding control group on novelty day).

**Figure 6 ijms-24-00586-f006:**
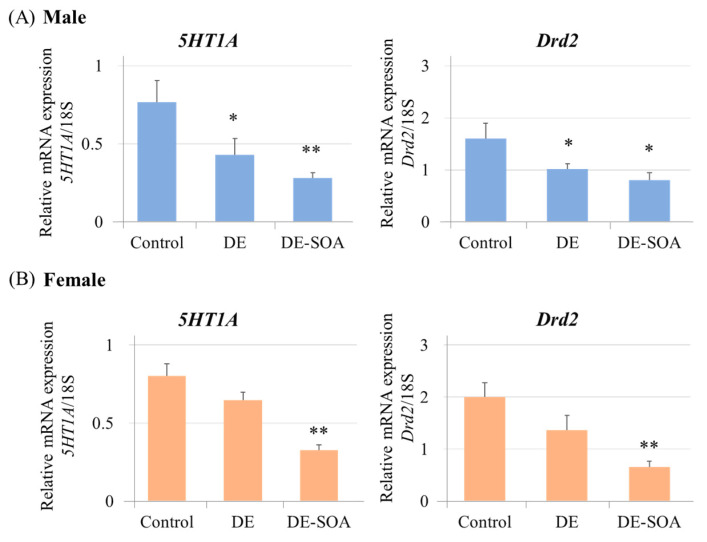
Messenger RNA expression levels of neurotransmitters in the hippocampus. Serotonin receptor (*5HT1A*) and dopamine receptor (*Drd2*) in the hippocampus of 10-week-old (**A**) male and (**B**) female rats of the control and the DE- or DE-SOA-exposed groups. (*n* = 10, ** *p* < 0.01, * *p* < 0.05 vs. control).

**Figure 7 ijms-24-00586-f007:**
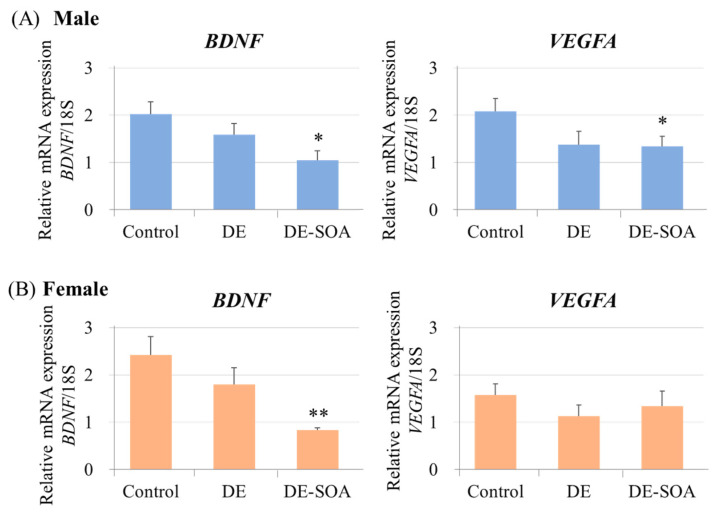
Messenger RNA expression levels of neurotrophins in the hippocampus. *BDNF* and *VEGFA* in the hippocampus of 10-week-old (**A**) male and (**B**) female rats of the control and the DE- or DE-SOA-exposed groups. (*n* = 10, ** *p* < 0.01, * *p* < 0.05 vs. control).

**Figure 8 ijms-24-00586-f008:**
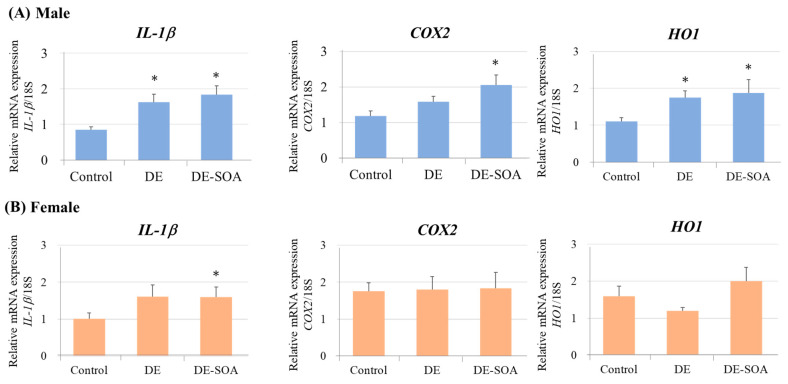
Messenger RNA expression levels of inflammatory and oxidative stress markers in the hippocampus. *IL-1β*, *COX2* and *HO1* mRNAs in the hippocampus of 10-week-old (**A**) male and (**B**) female rats of the control and the DE- or DE-SOA-exposed groups. (*n* = 10, * *p* < 0.05 vs. control).

**Figure 9 ijms-24-00586-f009:**
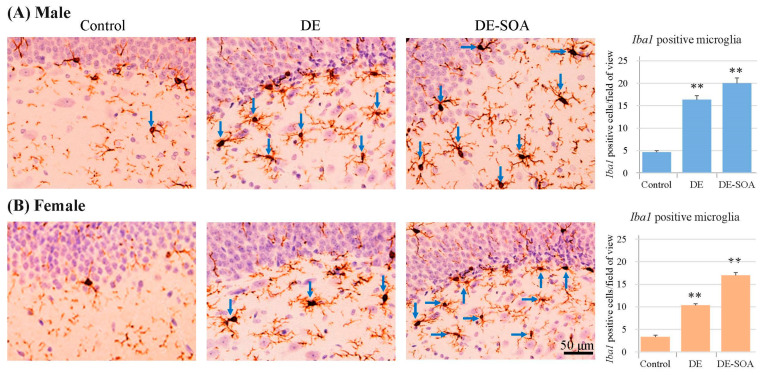
Representative photomicrographs of the dentate gyrus of the hippocampus. Microglia marker *Iba1* immunoreactivity in 10-week-old (**A**) male and (**B**) female rats of the control and the DE- or DE-SOA-exposed groups. Blue arrows indicate activated microglia. (Scale bar = 50 μm). Number of *Iba1*-positive cells located within the DG area of the hippocampus was quantified in a high-power field under microscopy (histogram). (*n* = 5, ** *p* < 0.01 vs. control).

**Figure 10 ijms-24-00586-f010:**
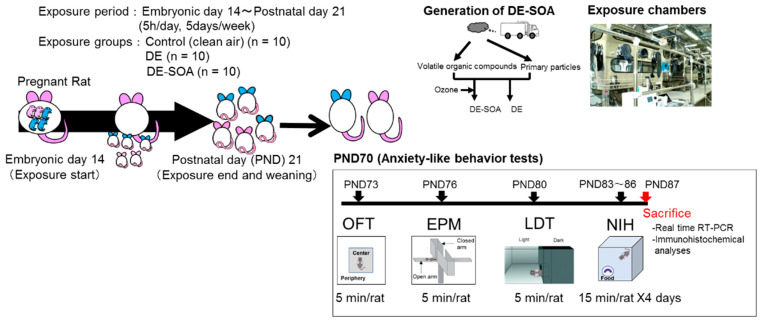
Experimental protocol. Pregnant Sprague Dawley rats (GD 8) were exposed to clean air (control, *n* = 8), DE (*n* = 8), and DE-SOA (*n* = 8) from GD 14 to PND 21 in whole-body exposure chambers. 10 male and 10 female rats from 3 groups (total 60) were used for the anxiety-like behavior tests, 5 male and 5 female rats from 3 groups (total 15) were used for immunohistochemical analyses. Chronological order of anxiety-like behavior tests was expressed.

**Table 1 ijms-24-00586-t001:** Summarized results of comparison of anxiety-like behavioral tests between DE and DE-SOA in male and female rats.

Behavioral Test	Parameters	DE (Compared to the Control)		DE-SOA (Compared to the Control)		Male & Female Difference	DE & DE-SOA Difference
		Male	Female	Male	Female		
OFT	Center entries	↓	↓↓	↓↓	↓↓	No	No
	Center time	↓↓	↓↓	↓↓	↓↓	No	No
EPM	Open arm entries	↓	↓↓	↓↓	↓↓	No	No
	Open arm time	—	—	—	—	No	—
LDT	Light entries	—	↑↑	—	—	Yes	Yes
	Light time	—	↑	—	—	Yes	Yes
NIH	Latency to approach food	↑	↑	↑	—	Yes	Yes
	Latency to eat food	↑	↑	↑	—	Yes	Yes

↓ Significantly decreased; ↑ Significantly increased.

**Table 2 ijms-24-00586-t002:** Summarized results of comparison of expression of molecular markers between DE and DE-SOA in male and female rats.

Molecular Markers	Parameters	DE (Compared to the Control)		DE-SOA (Compared to the Control)		Male & Female Difference	DE & DE-SOA Difference
		Male	Female	Male	Female		
Neurotransmitters	*5HT1A*	↓	—	↓↓	↓↓	Yes	No
	*Drd2*	↓	—	↓	↓↓	Yes	No
Neurotrophins	*BDNF*	—	↓	↓↓	↓↓	No	No
	*VEGFA*	—	—	↓	—	No	No
Proinflammatory cytokines	*IL-1β*	↑	—	—	↑	Yes	No
	*COX2*	—	—	↑	—	Yes	No
Oxidative stress marker	*HO1*	↑	—	↑	—	Yes	No
Microglia marker	*Iba1*	↑↑	↑↑	↑↑	↑↑	No	No

↓ Significantly decreased; ↑ Significantly increased.

**Table 3 ijms-24-00586-t003:** Characteristics of diesel exhaust particles and gaseous compounds in exposure chamber.

	Diesel Exhaust Particles			Temperature	Relative Humidity		
	Size (nm)	Particle Number (cm^−3^)	Concentration (mg/m^3^)	(°C)	(%)	EC/OC	WSOC/OC
Clean air	—	0.87 ± 0.57	13.2 ± 2.78	23.58 ± 0.27	48.07 ± 0.77	0.15 ± 0.06	0.03 ± 0.04
DE-SOA	24.45 ± 1.21	2.74 × 10⁶ ± 8.69 × 10⁴	118.23 ± 31.17	23.76 ± 0.19	48.4 ± 0.85	0.38 ± 0.03	0.11 ± 0.05
DE	22.69 ± 1.47	2.85 × 10⁶ ± 6.10 × 10⁴	101.9 ± 19.46	22.97± 0.21	49.32 ± 0.87	0.36 ± 0.03	0.17 ± 0.11
**Gaseous Compounds**							
	**CO (ppm)**	**SO_2_ (ppm)**	**NO_x_ (ppm)**	**NO_2_ (ppm)**	**NO (ppm)**	**O_3_ (ppm)**	**CO_2_ (%)**
Clean air	0.21 ± 0.04	0.00 ± 0.00	0.00 ± 0.00	0.00 ± 0.00	0.00 ± 0.00	—	0.05 ± 0.00
DE-SOA	2.51 ± 0.07	0.00 ± 0.00	1.14 ± 0.03	0.9 ± 0.03	0.15 ± 0.03	0.07 ± 0.00	0.07 ± 0.00
DE	2.52 ± 0.07	0.01 ± 0.00	1.21 ± 0.03	0.43 ± 0.02	0.78 ± 0.02	—	0.07 ± 0.00

Data were expressed as mean ± SD.

## Data Availability

All data generated or analyzed during this study are available from the corresponding author on reasonable request.

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
