# Peer review of "Early-Life Exposure to Traffic-Related Air Pollutants Induced Anxiety-like Behaviors in Rats via Neurotransmitters and Neurotrophic Factors"

_ijms, 2022, doi:10.3390/ijms24010586_

Round 1

Reviewer 1 Report

The manuscript entitled “Early-life Exposure to Traffic-related Air Pollutants induced Anxiety-like Behaviors in Rats via Neurotransmitters and Neurotrophic factors” shows findings on the association between early-life exposure to traffic-related air pollutants and the onset of anxiety-like behaviors in rats. The topic of the manuscript is very interesting but there are some questions regarding design and interpretation that decrease enthusiasm for the manuscript in its present form.  The section of introduction of the abstract with the results of its own previous experiments is confusing. Especially since this manuscript is primarily about the anxiety-causing effects of traffic-related air pollutants, not the induction of autism-like symptoms. It is important to mention this effect, but paraphrased. The authors should rewrite this section. In the results section of abstract, the authors should better show the meaning of the results and the significance of the research.

The section of the Introduction is short and does not include relevant information on the relationship between traffic-related air pollutant exposure and anxiety, there are no literature references.  Furthermore, it does not include information on the examined parameters, the levels of neurotransmitter receptors and neurotrophin associated with anxiety. The section of Results includes many poorly interpreted information. The sections of Material and Methods are well described with enough and wide information. The section of Discussion and Conclusion should be rewritten according to corrections. In general, the Conclusion seems to be vague and short. I recommend the following order: Abstract, Introduction, Materials and Methods, Results, Discussion, and Conclusion. However, the manuscript is poorly written, and in my opinion, it could be often benefit from significant reconsideration and reorganization. Furthermore, there are several minor experimental concerns that warrant attention.

However, I have some specific questions to this work:

P.3 line 82-88: “The open field test is basically used to assess anxiety in rodent. The male and female rats exposed to DE and DE-SOA groups showed a significantly decreased number of entries to the center and time stay in the center compared to the control group (p < 0.01; p < 84 0.05, Figure 2). Locomotor activity was not different between the control and exposure groups. However, grooming and defecation were increased in DE or DE-SOA exposed male and female rats. Our findings indicate that DE or DE-SOA exposure during perinatal period may induce anxiety-like behavior in male and female rats.”

v I have several problems with the analysis and evaluation of open field test.

1, “The open field test is basically used to assess anxiety in rodent.”

v This statement is true, however, the test is a simple assessment test also used to determine the level of spontaneous motor activity and exploratory behavior of rodents.

2, “The male and female rats exposed to DE and DE-SOA groups showed a significantly decreased number of entries to the center and time stay in the center compared to the control group (p < 0.01; p < 84 0.05, Figure 2).”

v According to me these data are not interpretable in the present form. Figure 2 shows that the number of entries is app. 2.5 in male control group, and app. 1 in male DE group during 5 min test.  Can this be evaluated as a significant reduction “in number of entries”?

3, “Locomotor activity was not different between the control and exposure groups. However, grooming and defecation were increased in DE or DE-SOA exposed male and female rats.”

v The authors note that no differences were found in locomotor activity between the groups. Please include these data in the manuscript. Is the “number of crossing” was analyzed in the open field test?

4, “However, grooming and defecation were increased in DE or DE-SOA exposed male and female rats.”

v It would be better to publish this information accurately. Were these parameters significantly increased? And other parameters such a rearing? The measurement of the freezing behavior and the number of rearing for example might reinforce the conclusions. In section 4.3.1 there is no mention this parameter: defecation?

5, “Our findings indicate that DE or DE-SOA exposure during perinatal period may induce anxiety-like behavior in male and female rats.” v This claim is not correct based on the above described results.

P3 line 94.: The elevated plus maze test was performed to assess the stress level of the mice.

v the use of “the rats or rodents” instead of “the mice”.

P4 line 105: 2.2.3. Light dark trtansition-spelling error: transition. 109-110: “These findings indicate that DE exposure may not induce some part of anxiety behavior in female rats.”

v The following phrase is not clear: some part of anxiety. Another wording is required. I suggest a more precise wording.

P6 line 149-151: “These findings indicate that 149 perinatal exposure to DE-SOA may induce reduction of BDNF levels in male and female rats in sex-dependent manner.”

v I don’t agree with this claim. You should rewrite it.

P8 line 185-187: The phrase in the Discussion (p.8): “DE-SOA induces (1) anxiety-like behaviors such as decreased entry to the center and time stay in the center in open field test, decreased open arm entry and open arm time in elevated plus maze test” is not correct.

v As described in 2.2 section and according to Figure 3, this conclusion is incorrect. It requires corrections. Furthermore, 2.1 Assessment of overall toxicity is not mentioned in section of Discussion.

P11 line 329-330: “the concentrations of DE and DE-SOA were 101 ± 9 μg/m3 and 118 ± 23 μg/m3, respectively.”

v Were these concentration values determined based on literature data? Furthermore, these data are not same as described in abstract.

P11 line 364-365: On test day, 364 animals were moved to testing room 30 min before experiment.

v In behavioral tests, the conditions of the experiment must be equal. For all behavioral tests, were the animals placed in the testing room 30 min before the experiment on the experimental day? Apart from the tested animal, were there other animals in the experimental room during the experiment?

There are minor points which need to revise.

P7 line 170: Microglial marker Iba1 was used to examine the activation of microglia in the hippo: “was used to examine” need to change italic letters to normal letters.

I suggest use of abbreviation of mentioned behavior tests such as I have found “Novelty-induced hypophagia” NIH test P9 line 223.

P1 line 37: the use of object instead of obejct

P1 line 74: the use of weight instead of weigh

Author Response

Responses to the Reviewer#1 comments

The manuscript entitled “Early-life Exposure to Traffic-related Air Pollutants induced Anxiety-like Behaviors in Rats via Neurotransmitters and Neurotrophic factors” shows findings on the association between early-life exposure to traffic-related air pollutants and the onset of anxiety-like behaviors in rats. The topic of the manuscript is very interesting but there are some questions regarding design and interpretation that decrease enthusiasm for the manuscript in its present form.  The section of introduction of the abstract with the results of its own previous experiments is confusing. Especially since this manuscript is primarily about the anxiety-causing effects of traffic-related air pollutants, not the induction of autism-like symptoms. It is important to mention this effect, but paraphrased. The authors should rewrite this section. In the results section of abstract, the authors should better show the meaning of the results and the significance of the research.

The section of the Introduction is short and does not include relevant information on the relationship between traffic-related air pollutant exposure and anxiety, there are no literature references.  Furthermore, it does not include information on the examined parameters, the levels of neurotransmitter receptors and neurotrophin associated with anxiety. The section of Results includes many poorly interpreted information. The sections of Material and Methods are well described with enough and wide information. The section of Discussion and Conclusion should be rewritten according to corrections. In general, the Conclusion seems to be vague and short. I recommend the following order: Abstract, Introduction, Materials and Methods, Results, Discussion, and Conclusion. However, the manuscript is poorly written, and in my opinion, it could be often benefit from significant reconsideration and reorganization. Furthermore, there are several minor experimental concerns that warrant attention.

However, I have some specific questions to this work:

P.3 line 82-88: “The open field test is basically used to assess anxiety in rodent. The male and female rats exposed to DE and DE-SOA groups showed a significantly decreased number of entries to the center and time stay in the center compared to the control group (p < 0.01; p < 84 0.05, Figure 2). Locomotor activity was not different between the control and exposure groups. However, grooming and defecation were increased in DE or DE-SOA exposed male and female rats. Our findings indicate that DE or DE-SOA exposure during perinatal period may induce anxiety-like behavior in male and female rats.”

v I have several problems with the analysis and evaluation of open field test.

1, “The open field test is basically used to assess anxiety in rodent.”

v This statement is true, however, the test is a simple assessment test also used to determine the level of spontaneous motor activity and exploratory behavior of rodents.

     As commented the Reviewer, we re-wrote this sentence to “The open field test is an experimental test used to determine general locomotor activity levels, anxiety, and exploratory behavior of rodents in scientific research.” in our revised manuscript.

2, “The male and female rats exposed to DE and DE-SOA groups showed a significantly decreased number of entries to the center and time stay in the center compared to the control group (p < 0.01; p < 0.05, Figure 2).”

v According to me these data are not interpretable in the present form. Figure 2 shows that the number of entries is app. 2.5 in male control group, and app. 1 in male DE group during 5 min test.  Can this be evaluated as a significant reduction “in number of entries”?

      According to our results, number of entries was; control = 2.6 ± 0.3, DE = 1.2 ± 0.2 and DE-SOA = 0.9 ± 0.2 times per 5 min for 10 rats per group. It was statistically significant reduction of center entry time in open field test.

3, “Locomotor activity was not different between the control and exposure groups. However, grooming and defecation were increased in DE or DE-SOA exposed male and female rats.”

The authors note that no differences were found in locomotor activity between the groups. Please include these data in the manuscript. Is the “number of crossing” was analyzed in the open field test?

      We revised the word “locomotor activity” to “distance travel” in our revised manuscript. Unfortunately, number of crossing was not measured in the open field test.

4, “However, grooming and defecation were increased in DE or DE-SOA exposed male and female rats.”

v It would be better to publish this information accurately. Were these parameters significantly increased? And other parameters such a rearing? The measurement of the freezing behavior and the number of rearing for example might reinforce the conclusions. In section 4.3.1 there is no mention this parameter: defecation?

      We provided grooming (wash the face and body), rearing (standing on hindlimbs) and defecation (fecal pellets) data in small table in Fig. 2 of our revised manuscript.

5, “Our findings indicate that DE or DE-SOA exposure during perinatal period may induce anxiety-like behavior in male and female rats.” v This claim is not correct based on the above described results.

    We corrected that sentence to Our findings indicate that DE or DE-SOA exposure during perinatal period may induce emotional insecurity and restlessness in male and female rats.” In our revised manuscript.

P3 line 94.: The elevated plus maze test was performed to assess the stress level of the mice.

v the use of “the rats or rodents” instead of “the mice”.

   Thank you for pointing out our mistake, we corrected “the mice” to “the rats or rodents” in our revised manuscript.

P4 line 105: 2.2.3. Light dark trtansition-spelling error: transition. 109-110: “These findings indicate that DE exposure may not induce some part of anxiety behavior in female rats.”

   We corrected to “transition”. We re-wrote the sentence to “These findings indicate that DE exposure may affect some part of anxiety behavior in sex specific manner.” in our revised manuscript.

The following phrase is not clear: some part of anxiety. Another wording is required. I suggest a more precise wording.

P6 line 149-151: “These findings indicate that perinatal exposure to DE-SOA may induce reduction of BDNF levels in male and female rats in sex-dependent manner.”

v I don’t agree with this claim. You should rewrite it.

      We would like to thank the Reviewer for pointing out our mistake. We corrected that sentence to “These findings indicate that perinatal exposure to DE-SOA may induce reduction of BDNF levels in male and female rats.” in our revised manuscript.

P8 line 185-187: The phrase in the Discussion (p.8): “DE-SOA induces (1) anxiety-like behaviors such as decreased entry to the center and time stay in the center in open field test, decreased open arm entry and open arm time in elevated plus maze test” is not correct.

As described in 2.2 section and according to Figure 3, this conclusion is incorrect. It requires corrections. Furthermore, 2.1 Assessment of overall toxicity is not mentioned in section of Discussion.

      We would like to thank the Reviewer for pointing out our mistake. We corrected that sentence to “The male and female rats exposed to DE or DE-SOA groups showed a significantly decreased open arms entry compared to the control group (p < 0.01, p < 0.05, Figure 3) but not significant in time spent in open arms.” in our revised manuscript.

P11 line 329-330: “the concentrations of DE and DE-SOA were 101 ± 9 μg/m3 and 118 ± 23 μg/m3, respectively.”

v Were these concentration values determined based on literature data? Furthermore, these data are not same as described in abstract.

      We are sorry for misunderstanding. Actually we expressed nearest digit from 101 ± 9 μg/m3 to 100 μg/m3 and 118 ± 23 μg/m3 to 120 μg/m3. We corrected same concentration in ABSTRACT of our newly revised manuscript.

      For the whole-body inhalation exposure for rodents, the concentrations were set at approximately same levels in the chambers for generation of nano-sized particle rich DE and DE-SOA in our Research Institute.

P11 line 364-365: On test day, animals were moved to testing room 30 min before experiment.

v In behavioral tests, the conditions of the experiment must be equal. For all behavioral tests, were the animals placed in the testing room 30 min before the experiment on the experimental day? Apart from the tested animal, were there other animals in the experimental room during the experiment?

  We are sorry for misunderstanding for that issue. All the animals were kept in the animal experimental room next to the behavior test room. Animals were placed in the testing room 30 min before the behavior test on the experimental day. Apart from the tested animal, other animals were kept in the animal experimental room during the whole experiment period. We provided this issue in Materials and Methods section in our revised manuscript.

There are minor points which need to revise.

P7 line 170: Microglial marker Iba1 was used to examine the activation of microglia in the hippo: “was used to examine” need to change italic letters to normal letters.

   As commented by the Reviewer, we corrected “was used to examine”(italic letters) to normal letters.

I suggest use of abbreviation of mentioned behavior tests such as I have found “Novelty-induced hypophagia” NIH test P9 line 223.

   As suggested by the Reviewer, we used abbreviation of behavior tests in our revised manuscript.

P1 line 37: the use of object instead of obejct

Thank you for pointing out spelling miss, we corrected to “object” in PX, line XX in our revised manuscript.

P1 line 74: the use of weight instead of weigh

   Thank you for pointing out spelling miss, we corrected to “weight” in PX, line XX in our revised manuscript.

Reviewer 2 Report

This paper appears interesting in the field, but in the present form, the manuscript appears descriptive, as a list of putative markers deregulated. In order to improve the general scientific quality for readers, a specific attention should be paid to mechanisms involved and putative pathways in brain cells leading to the monitored functional outcomes. Indirect correlations with the literature are not sufficient.

More precisely:

The introduction appears weak and insufficiently detailed concerning the developmental mechanisms involved in pathogenic related outcomes. Actually, authors should present the different phases of neurodevelopment in the light of the diesel exposure: early neural crests formation and putative pathologies, neurogenesis and brain sub-regions formation, synaptic connectivity and maturation before and after birth. Do all of them equally sensitive to diesel exposure? It is slightly mentioned in lines 41-44, but without reference and no details and/or mechanisms. This is highly relevant for this paper because the choice to expose dams between gestational day 14 and post-natal day 21. This point must be explain in the beginning of the methods and/or the discussion. For example, why not before gestational day 14? ...etc explain your choice and the relevance of your model.

ASD is insufficiently referenced lines 44-49.

The end of the introduction lines 50-61 is not sufficiently compared with other studies.

The relevance to study DE and DE-SOA , DE versus DE-SOA or other aspects is not sufficiently explained. What is DE-SOPA line 66 ?

Results: The scale of numerous graphs are not well chosen.

Methods:

Even if a lot of precautions were taken to analyze and standardize the gas generation and delivery, authors should explain their choices. Explain the relevance of DE and DESOA concentrations in the light of experimental necessity and physiological putative exposition for humans. Why 5 days a week for exposition? Is it possible to present a table indicating the gas composition for other scientists in the field?

As a general observation, it has been shown that cleaning behavioral apparatus using chemical agents and 70% ethanol could generate anxiety or at least modify the tested behavior by itself (personal experience and Bartolomé et al., 2017, Behav Brain Res; Hershey et al., 2018, J Am Assoc Lab Anim Sci). Did authors applied a delay between the use of ethanol and the successive runs ? It is mentioned only for the light dark test that the apparatus was dried after ethanol. We can suggest that authors propose a first paragraph in the method section before each test presentation, for general comments common to every tests.

It appears necessary to propose a chronological figure to present of all tests and experiments performed because we need the time-laps of tests, the age of offspring at the time of each test. It is possible that a test influence another.

Concerning the open-field, it is surprising to see that a square was designed in the center as a circle (?) of 25-cm diameter. Is it a square of 25-cm width x 25-cm length ? Was it a square or a circle ?

Concerning mRNA assay; it is mentioned that the prefrontal cortex was collected, but in the rest of the manuscript authors present and discuss for the hippocampus. It is a major problem.

Discussion

The manuscript needs a general revision for the English writing, and it is surprising that the presentation of the result is in the past rather than usually in the present with the discussion in the past.

In a mechanistic point of view, lines 234 to 251 are not clearly related to the present study since the amygdala-related functions exposed by the literature appear not really relevant to the present study focused on the hippocampus (or cortex, we don't know reading the method section).

Also, authors did not discuss the difference between DE and DE-SOA.

The putative difference between males and females are not well discussed.

Finally, the discussion appears as a list of well-known markers that numerous of them were already studied in related brain disorders. Authors should argue for alternative mechanistic hypothesis if they do not have other actors to focus on pathways in neurons or glial cells.

Author Response

Responses to the Reviewer#2 comments

This paper appears interesting in the field, but in the present form, the manuscript appears descriptive, as a list of putative markers deregulated. In order to improve the general scientific quality for readers, a specific attention should be paid to mechanisms involved and putative pathways in brain cells leading to the monitored functional outcomes. Indirect correlations with the literature are not sufficient.

More precisely:

The introduction appears weak and insufficiently detailed concerning the developmental mechanisms involved in pathogenic related outcomes. Actually, authors should present the different phases of neurodevelopment in the light of the diesel exposure: early neural crests formation and putative pathologies, neurogenesis and brain sub-regions formation, synaptic connectivity and maturation before and after birth. Do all of them equally sensitive to diesel exposure? It is slightly mentioned in lines 41-44, but without reference and no details and/or mechanisms. This is highly relevant for this paper because the choice to expose dams between gestational day 14 and post-natal day 21. This point must be explain in the beginning of the methods and/or the discussion. For example, why not before gestational day 14? ...etc explain your choice and the relevance of your model.

   As commented by the Reviewer, we re-wrote INTRODUCTION in our revised manuscript.

   As commented by the Reviewer, neural crest formation, neurogenesis and neuronal network programming occur in different developmental period. Regarding this issue, we added “In preliminary studies, our research group has investigated the critical period of exposure to nanosized diesel exhaust particles for brain functions and behaviors using different phase of developmental periods such as the whole gestational period, early, middle, and late gestational period, neonatal period, and perinatal period. We found that the perinatal period (GD 14 to PND21) was the most vulnerable period for the whole-body exposure to DE or DE-SOA. In addition, epidemiological studies have reported that exposure to air pollutants during prenatal and childhood periods was associated with neurodevelopmental and emotional disorders including ASD, ADHD, anxiety and depression (Forns et al., 2016; Yolton et al., 2019; Brunst et al., 2019). Thus, in this study, we have selected that perinatal period to examine the effect of developmental exposure to DE or DE-SOA on neuropsychiatric diseases especially anxiety-like behavior and neurological and immunological markers using rat models.” in DISCUSSION session of our revised manuscript.

ASD is insufficiently referenced lines 44-49.

   We deleted un-relevant ASD related references in our revised manuscript.

The end of the introduction lines 50-61 is not sufficiently compared with other studies.

    As suggested by the Reviewer, we deleted un-relevant sentences and references.

The relevance to study DE and DE-SOA, DE versus DE-SOA or other aspects is not sufficiently explained. What is DE-SOPA line 66 ?

   We added the summarized results of difference between DE and DE-SOA in male and female rats in Table 2 and 3 and discuss ed that issue in our revised manuscript

We corrected DE-SOPA to DE-SOA in our revised manuscript.

Results: The scale of numerous graphs are not well chosen.

   We corrected scale of graphs in Fig. 5, 6, 7,8 and 9 in our revised manuscript.

Methods:

Even if a lot of precautions were taken to analyze and standardize the gas generation and delivery, authors should explain their choices. Explain the relevance of DE and DESOA concentrations in the light of experimental necessity and physiological putative exposition for humans. Why 5 days a week for exposition? Is it possible to present a table indicating the gas composition for other scientists in the field?

   At the beginning of all series of experiments for developmental and adult exposure to DE or DE-SOA, our Research Institute considered schedule which closed to real world exposure scenario, e.g., for working pregnant mothers who work about 5h per day for 5 working days (weekdays).

   As commented by the Reviewer, we provided gas composition in Table 1 in our revised manuscript.

As a general observation, it has been shown that cleaning behavioral apparatus using chemical agents and 70% ethanol could generate anxiety or at least modify the tested behavior by itself (personal experience and Bartolomé et al., 2017, Behav Brain Res; Hershey et al., 2018, J Am Assoc Lab Anim Sci). Did authors applied a delay between the use of ethanol and the successive runs ? It is mentioned only for the light dark test that the apparatus was dried after ethanol. We can suggest that authors propose a first paragraph in the method section before each test presentation, for general comments common to every tests.

   We agree with the Reviewer that 70% ethanol can induce anxiety or influence the behavioral tests. We paid attention for this issue for every behavior test, and we made a sufficient interval between test rats and after cleaning with 70% ethanol, we used handy fan to remove the smell of ethanol. As commented by the Reviewer, we added this issue in the first paragraph of the Method section in our revised manuscript.

It appears necessary to propose a chronological figure to present of all tests and experiments performed because we need the time-laps of tests, the age of offspring at the time of each test. It is possible that a test influence another.

   As commented by the Reviewer, we added a chronological figure for all behavior test timeline in Figure 10 in our revised manuscript.

Concerning the open-field, it is surprising to see that a square was designed in the center as a circle (?) of 25-cm diameter. Is it a square of 25-cm width x 25-cm length ? Was it a square or a circle ?

   We thank the Reviewer for pointing out our mistake. We corrected as “a center square (25 cm X 25 cm) was outlined in red tape.” in our revised manuscript.

Concerning mRNA assay; it is mentioned that the prefrontal cortex was collected, but in the rest of the manuscript authors present and discuss for the hippocampus. It is a major problem.

    Actually, we collected the hippocampus for the present study. We apologize for our mistake. We corrected “the prefrontal cortex” to “the hippocampus” in our revised manuscript.

Discussion

The manuscript needs a general revision for the English writing, and it is surprising that the presentation of the result is in the past rather than usually in the present with the discussion in the past.

   Thank you for comments to improve our manuscript and we revised English writing in our revised manuscript.

In a mechanistic point of view, lines 234 to 251 are not clearly related to the present study since the amygdala-related functions exposed by the literature appear not really relevant to the present study focused on the hippocampus (or cortex, we don't know reading the method section).

   As suggested by the Reviewer, we deleted that two paragraph and added relevant literature in our revised manuscript as follows,

Actually, we collected the hippocampus for the present study. We apologize for our mistake. We corrected “the prefrontal cortex” to “the hippocampus” in our revised manuscript.

Also, authors did not discuss the difference between DE and DE-SOA.

   As commented by the Reviewer, we add table 2 and 3 and also discussed that issue as “The significant difference of anxiety-like behaviors and neurological and immunological marker mRNAs between the control and DE-SOA groups were observed, but not between DE and DE-SOA groups” in our revised manuscript.

The putative difference between males and females are not well discussed.

   Regarding that issue, we have already discussed in second last paragraph of discussion section as follows; In this study, some sex-dependent manner of anxiety-like behavior and neurological markers were observed. The exact etiology was not known, but female may possess some protective activities than in male. It has been reported that estrogen exerts protective actions by suppressing the neurotoxic stimulus or increasing the resilience of the brain to a given injury [49]. Moreover, paraoxonase-2, is a mitochondrial enzyme, possesses potent antioxidant and anti-inflammatory properties, and its expression in female monkeys was significantly higher than in male monkeys in multiple brain regions [50].

Finally, the discussion appears as a list of well-known markers that numerous of them were already studied in related brain disorders. Authors should argue for alternative mechanistic hypothesis if they do not have other actors to focus on pathways in neurons or glial cells.

   As suggested by the Reviewer, we re-write DISCUSSION section in our revised manuscript.

Round 2

Reviewer 1 Report

I find your manuscript with these modifications acceptable for publication.

Author Response

Response to the Reviewer#1’s comment (R2)

  • I find your manuscript with these modifications acceptable for publication.

We really appreciate your invaluable comments and suggestion to improve our manuscript.

Reviewer 2 Report

Introduction

The introduction is reformulated but presents a list of indications in the field rather than a scientific argumentation in the favor of the study, except the previous publication of the group.

line 78 Ji et al., 2022 appears not well cited

paragraph lines 90 to 100 gives the same information than the beginning of the introduction, must be reformulated.

Authors switch from human to animal studies in the whole introduction; should be clarified to present a scientific line instead a list of correlations.

Discussion

Since the study is focused on hippocampus only, it appears restrictive in term of tissue analysis compared to the global aim of anxiety. Also, serotonin and dopamine receptors are relevant for hippocampus functions, but they are not typically associated to the role of hippocampus in anxiety. Even the presented references indicate a larger influence of these two neuro-receptors in the whole brain concerning anxiety. Authors should discuss more this point instead of linking their results obtained on the hippocampus and a global effect.

Why authors did not discuss the difference between DE and DE-SOA? If a difference was made in the experimental approach, a discussion for that point should be produced. Do DE and DESOA behave differently when entering a brain ? Are the behavioral consequences different? Could we have a rapid presentation of the literature for that ? Especially because authors discuss sometimes with a difference between controls and DE rats, and sometimes between controls and DESOA-rats. Finally, it's confusing. The poor presentation lines 350-352 appears weak for that. If there is no typical difference, present it and explain your initial choice.

In a general point of view, the experimental design and the method appear good, but (1) the choice to focus only on the hippocampus while measuring more global markers in a molecular point of view is a shame, and (2) the discussion of results gives too direct correlations between molecular markers one by one and a global psychological feeling such as anxiety, without presenting any putative mechanistic deregulations. So, readers would take the results as descriptive only for the hippocampus of exposed rats.

Author Response

Responses to the Reviewer#2’s comments (R2)

   We would like to thank the Reviewer#2 for invaluable comments and suggestion for improvement of our manuscript. We have responded point by point to the Reviewer’s comments.

Introduction

The introduction is reformulated but presents a list of indications in the field rather than a scientific argumentation in the favor of the study, except the previous publication of the group.

  • line 78 Ji et al., 2022 appears not well cited

We would like to thank the Reviewer for pointing out our mistake. We deleted that in our revised manuscript.

  • paragraph lines 90 to 100 gives the same information than the beginning of the introduction, must be reformulated.

We deleted the overlapped information in our revised manuscript.

  • Authors switch from human to animal studies in the whole introduction; should be clarified to present a scientific line instead a list of correlations.

We edited and added relevant scientific information in our revised manuscript.

Discussion

Since the study is focused on hippocampus only, it appears restrictive in term of tissue analysis compared to the global aim of anxiety. Also, serotonin and dopamine receptors are relevant for hippocampus functions, but they are not typically associated to the role of hippocampus in anxiety. Even the presented references indicate a larger influence of these two neuro-receptors in the whole brain concerning anxiety. Authors should discuss more this point instead of linking their results obtained on the hippocampus and a global effect.

   As commented buy the reviewer, we added why we selected the hippocampus for this study in the introduction session in our revised manuscript as follows,

Hippocampus is the important brain area for cognitive functions such as episodic memory and spatial navigation. In addition, it is also involved in the pathogenesis of mood and anxiety disorders. The dorsal hippocampus contributes to cognitive functions, and the ventral hippocampus modulates emotional regulation (Fanselow and Dong, 2010; Strange et al., 2014).  The hippocampus, amygdala and prefrontal cortex are the brain areas which plays a major role in emotional behaviors and functions. The neural connections projecting from ventral hippocampus to the prefrontal cortex are unidirectional. The ventral hippocampus also has bidirectional connections with the amygdala, and the amygdala has bidirectional connections with the prefrontal cortex. Because the amygdala and hippocampus are known to be involved in emotional and contextual memory processing, stress likely contributes to the dysregulation of these functions. Using freely moving calcium imaging and optogenetics, it was reported that optogenetic activation of ventral hippocampal terminals in lateral hypothalamus but not basal amygdala increased anxiety and avoidance. Ventral hippocampal area is enriched in anxiety cells thus, the hippocampus can rapidly influence innate anxiety behavior directly via ventral hippocampus-lateral hypothalamus pathway (Jimenez JC et al., 2018). Transgenic animals with impaired hippocampal neurogenesis exhibit a significantly increased anxiety-like behaviors (Revest et al., 2009). Taken together, the hippocampus is the essential brain region for anxiety-like behaviors when expose to environmental insults. Thus, in the present study, the hippocampus was selected to examine the environmental pollutant-induced anxiety-like behaviors in rats.

Why authors did not discuss the difference between DE and DE-SOA? If a difference was made in the experimental approach, a discussion for that point should be produced. Do DE and DESOA behave differently when entering a brain ? Are the behavioral consequences different? Could we have a rapid presentation of the literature for that ? Especially because authors discuss sometimes with a difference between controls and DE rats, and sometimes between controls and DESOA-rats. Finally, it's confusing. The poor presentation lines 350-352 appears weak for that. If there is no typical difference, present it and explain your initial choice.

In a general point of view, the experimental design and the method appear good, but (1) the choice to focus only on the hippocampus while measuring more global markers in a molecular point of view is a shame, and (2) the discussion of results gives too direct correlations between molecular markers one by one and a global psychological feeling such as anxiety, without presenting any putative mechanistic deregulations. So, readers would take the results as descriptive only for the hippocampus of exposed rats.

Regarding the difference between DE and DE-SOA is very complicated to explain as shown in the Table 2 and 3. The effects of DE or DE-SOA are brain region-specific, and it should be better to examine the related brain regions simultaneously. Hippocampal volume, neurogenesis, apoptosis, autophagy are the other mechanisms of contributing factors for hippocampus-dependent anxiety-like behaviors following air-pollutant exposure. The impact of exposure to DE-SOA appears greater than DE, but not statistically significant. DE is precursor of DE-SOA and adding ozone to oxidize to form DE-SOA. Moreover, we could not mention exactly the differences between DE and DE-SOA for translocation to brain, the effects on the anxiety-like behavior and molecular markers in the rat model with or without OVA immunization. The limitation of this study was that only hippocampus was examined. We have a plan to examine the constituents of DE or DE-SOA and translocation to the brain and the different effects on other behavior and molecular markers in future studies.

   We added that issue in our revised manuscript.

Round 3

Reviewer 2 Report

This last version of the manuscript jumps to a higher quality. Nevertheless, only descriptive correlations between DE and different markers are presented. Almost no hypothesis for putative intracellular mechanisms is proposed or discussed. Indeed, no correction or treatment for exposed persons could emerged, except the recommendation for a non-exposition.